# Communicating Natural Programs to Humans and Machines

**Samuel Acquaviva**[*]
MIT

**Yewen Pu**[*]
Autodesk Research

**Marta Kryven** [†]
MIT

**Theodoros Sechopoulos** [†]
MIT

**Catherine Wong** [†]
MIT

**Gabrielle E Ecanow**
MIT

**Maxwell Nye**
MIT

**Michael Henry Tessler**
MIT

**Joshua B. Tenenbaum**
MIT

## Abstract

The Abstraction and Reasoning Corpus (ARC) is a set of procedural tasks that tests an agent's ability to flexibly solve novel problems. While most ARC tasks are easy for humans, they are challenging for state-of-the-art AI. What makes building intelligent systems that can generalize to novel situations such as ARC difficult? We posit that the answer might be found by studying the difference of *language*: While humans readily generate and interpret instructions in a general language, computer systems are shackled to a narrow domain-specific language that they can precisely execute. We present LARC, the *Language-complete ARC*: a collection of natural language descriptions by a group of human participants who instruct each other on how to solve ARC tasks using language alone, which contains successful instructions for 88% of the ARC tasks. We analyze the collected instructions as 'natural programs', finding that while they resemble computer programs, they are distinct in two ways: First, they contain a wide range of primitives; Second, they frequently leverage communicative strategies beyond directly executable codes. We demonstrate that these two distinctions prevent current program synthesis techniques from leveraging LARC to its full potential, and give concrete suggestions on how to build the next-generation program synthesizers.

## 1 Introduction

Humans solve a range of procedural tasks such as cooking, tying shoes, and programming. Although current AI systems achieve super-human proficiency at certain narrowly specified tasks [1,2], their reasoning is domain-specific and fails to generalize to novel situations [3]. The *Abstraction and Reasoning Corpus* (ARC) introduced by [4] presents a set of procedural tasks constructed expressly to benchmark fundamental capacities associated with human general intelligence, including abstraction, generalization, object categories, and procedural analogies [3,5–10]. Specifically, ARC requires one to infer a procedure consistent with a small number of abstract input-output examples and apply it to a new input to generate an unseen answer, see Figure 1.

How do we build systems that are capable of solving general, procedural tasks such as ARC? Traditional approaches of program synthesis [11–14] and semantic parsing [15–20] assume the tasks are **DSL-closed** – for any task, there exists a program, written in a predefined *Domain Specific Language* (DSL), that solves the task. The ARC benchmark is uniquely *designed* to be **DSL-open**

---

[*]and [†] denote equal contributions

36th Conference on Neural Information Processing Systems (NeurIPS 2022) Track on Datasets and Benchmarks.

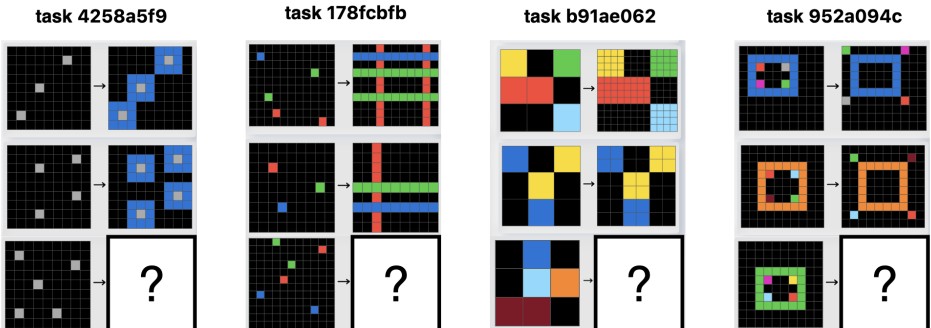

Figure 1: Four ARC tasks, the goal is to correctly infer the unseen output from the given examples.

– it does not come with a predefined DSL capable of representing its tasks intuitively. This is both reasonable – most real life tasks, such as cooking and assembling furniture, are DSL-open – and challenging – how can one build an intelligent system that can solve tasks from few examples without a DSL? To illustrate, what might a DSL that would allow one to program all the ARC tasks in Figure 1 look like? This question is difficult to answer: a recent Kaggle competition found that the best AI systems solve at most 20% of the tasks, while [21] found that most humans easily solve over 80% [2].

Given that humans greatly outperform the best AI systems in solving ARC tasks, studying the human's cognitive processes (for instance, which set of concepts do human use to represent these tasks?) can shed light on how to build similarly intelligent systems. As these thought processes are not observable directly, we study **natural programs** – instructions that humans give to each other, as a window into these latent cognitive processes. Like computer programs, these instructions can be reliably interpreted (by another human) to produce the intended output. Unlike computer programs, which must be stated in a specific style, natural programs can be stated in any form – such as verbal instructions or input-output examples – as long as another human can execute them. In this work, we study a particular form of natural programs, that of *natural language instructions*. We show that analyzing these natural programs – with explicit comparisons to computer programs – can both shed light on how humans communicate and interpret procedures [22–25] and inform how one may build AI systems for challenging, DSL-open domains such as ARC.

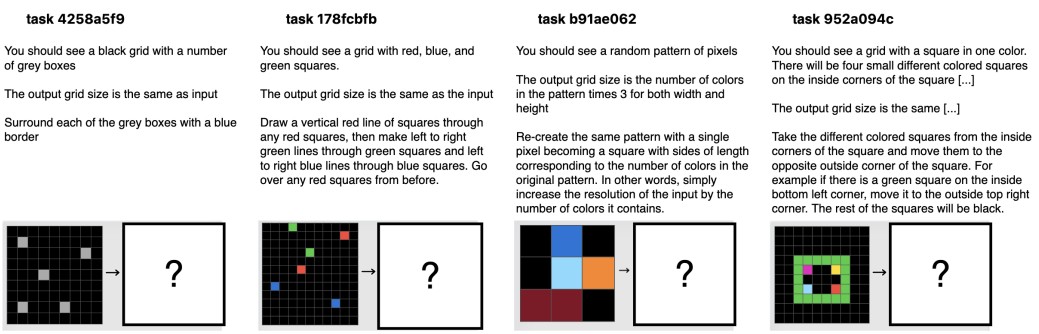

Figure 2: Four LARC tasks, corresponding to those of Figure 1. The goal is to produce the correct output given *only* the language instructions. 88% of the ARC tasks can be communicated this way. What are some of the communicative strategies used by humans here?

We present the **Language-complete Abstraction and Reasoning Corpus** (LARC) [3] of natural language instructions elicited from a two-player communication game, where 88% of the ARC tasks can be successfully communicated. LARC tasks are **language-complete**: The successful instructions contain all the relevant information, even in absence of the original input-output examples (see Figure 2). This is important in several ways: First, one can use LARC to study how humans use language to communicate abstract procedures, as humans clearly have the capacity to both *generate* and *execute*

---

[2]Humans were evaluated on a subset of the training tasks; the Kaggle competition used a private test set.
[3]https://github.com/samacqua/LARC

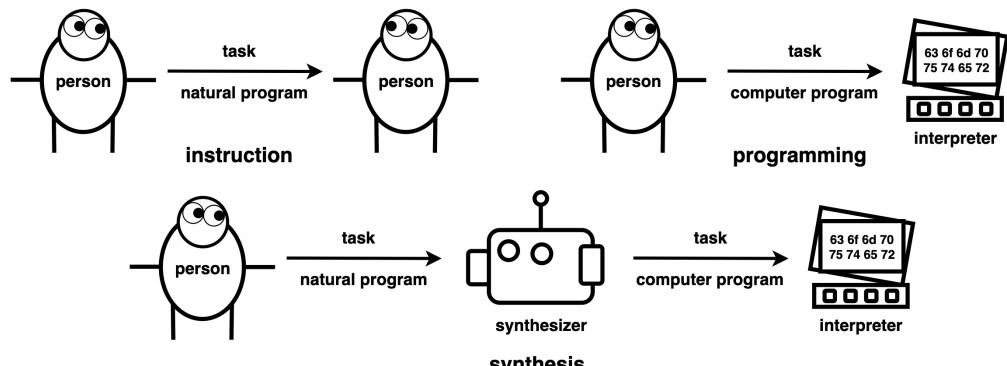

Figure 3: Three kinds of "programs": instruction (top-left), programming (top-right), synthesis (bot).

these natural programs; Second, one can directly see what concepts an intelligent system must be aware of (such as colors and numbers); Third, as people readily generate natural programs, studying them will provide insights on building interactive systems.

We perform **linguistic analysis** on LARC, finding that humans readily leverage algorithmic concepts without being explicitly instructed to do so. These concepts range from domain general ones, such as loops, to domain-specific concepts such as flood-fill. However, natural programs in LARC are distinct from typical computer programs in two ways: (1) natural programs use a much wider range of concepts compared to a typical DSL; (2) natural programs contain clarifications and validations in greater quantity than directly executable procedures. We apply standard **program synthesis** algorithms on LARC, finding that while existing approaches can benefit from the additional language annotations, the two aforementioned distinctions pose significant challenges to standard program synthesis approaches. We conclude by providing concrete suggestions on how to build the next generation program synthesizers.

## 2 Communicating and Interpreting Programs

In programming, a *programmer* constructs a *program* in a suitable language, which is then executed on an *interpreter*, producing a behaviour. For instance, a person can *instruct* another person to carry out a certain task (Fig. 3 top-left), or directly *program* a machine to solve tasks using code (Fig. 3 top-right). A program synthesizer takes in an instruction, and reformulates it as code, insulating the person from the programming process (Fig. 3 bot). We treat all three as acts of *programming*.

How do we build systems that can be communicated naturally to solve challenging tasks? Typically, one follows a "DSL-first" approach, where one first defines a programming language and builds a corresponding interpreter capable of executing programs written in this language. Then, one naturalizes the initial DSL using synthesis, allowing end-users to describe tasks using natural language [15–18, 26, 27], or by giving examples [12, 13, 28]. While this DSL-first workflow has yielded impressive results, the DSL itself is also a single point of failure. It is difficult to design DSL with the right *scope*, so that it both expressive and non-redundant [29–31]. One must ensure that the DSL *aligns* reasonably to human instructions [32, 33], while simultaneously being *efficient* when used by the synthesizer [12, 34]. These challenges may explain why ARC, and other DSL-open domains (where procedural tasks are given *in the absence* of a narrow DSL), are difficult to tackle.

In this work, we adopt the Wizard-of-Oz approach [35–37] by using a human as an interpreter of natural language instructions (Fig 3 top-left). We define a **natural program** as instructions constructed by a person that can be interpreted by another person to produce a specific output. This program is *natural*–it can be understood by speakers of the language[4] without a prior consensus–but behaves as a *program*, in that it produces a definitive output, which can be unambiguously checked for correctness. For instance, the original ARC [4] tasks are natural programs: Given a program consisting of input-output examples, a fellow human can readily interpret this program to produce an

---

[4]language here is to be understood loosely as any medium of communication between people

output on a new input, which can be checked for correctness. By starting with (linguistic) natural programs, one can directly observe the set of concepts and strategies necessary to master a domain (such as ARC), without committing to a specific interpreter.

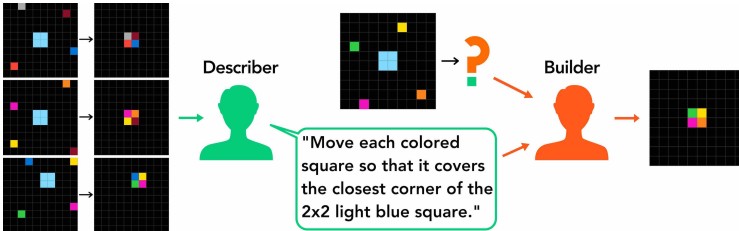

Figure 4: a **describer** instructs a **builder** how to solve an ARC task using a natural program

## 3 LARC: Language-complete Abstraction and Reasoning Corpus

We present a dataset that augments the original ARC tasks from [4] with *language-complete* instructions: they can be demonstrably interpreted by other humans to correctly produce the intended outputs without any additional contexts (i.e. in the absence of the original input-output examples). Thus, LARC tasks (Fig 2), like their counterparts in ARC, meet the definition of natural program while containing only natural language descriptions. To collect this dataset, we introduce a *communication game*: human describers produce linguistic instructions from the given input-output examples of ARC, these instructions are then interpreted by human builders (in the absence of the original input-output) on a new instance of the same task (Fig. 4). We deployed this experiment using a novel bandit algorithm to efficiently collect verifiable natural programs. The final dataset augments 88% of the original ARC tasks (354/400) with at least one verifiable *natural program* description that could be successfully interpreted by another human participant to solve the task. Fig. 5(C-D) shows the distribution of success rates for participants acting as describers and builders over time.

### 3.1 Human annotation details

We recruited 373 subjects via Amazon Mechanical Turk who were paid for 45 minutes of work. Fifty individuals were excluded for failing to complete the task, so the final analysis included 323 subjects. The study was approved by our institution's Institutional Review Board, did not collect personally identifiable information, and did not pose risks to participants. Subjects were paid $6.00 and a $0.25 bonus for every successful communication. Subjects averaged 5.5 communications, bringing their expected hourly wage to $9.83. For interface and consent form see Appendix A.2. [5]

### 3.2 Two-player communication game

For each task, a participant may be assigned one of two roles: a **describer** or a **builder**. The describer plays the role of a *human synthesizer*, who reformulates input-output examples (of ARC) to natural language descriptions. The builder plays a role of a *human interpreter*, who must construct the correct output on a new input without access to the original examples (Fig 4). The description is structured into three sections to incentivize consistency: (1) what the builder should expect to see in the input, (2) the output grid size, and (3) what the builder should do to create the output (Fig 2). After the description was submitted, we verify the describer's own understanding by asking them to build it, and discarding the submission if the describer fails. The describer was shown all previous verified descriptions for a task, alleviating challenge of solving the task from scratch. Builders construct/draw the output using actions defined in ARC, such as `paint(color,x,y)`, `copy/paste`, and `floodfill`. All drawing sequences are recorded and can be played back.

### 3.3 The Bandit Algorithm for Data Collection

Collecting valid linguistic natural programs requires significant human efforts: For each task (of varying difficulties), natural programs must first be *proposed* by a number of describers, and then

---

[5] see https://arxiv.org/abs/2106.07824 for full paper with appendix attached at the end

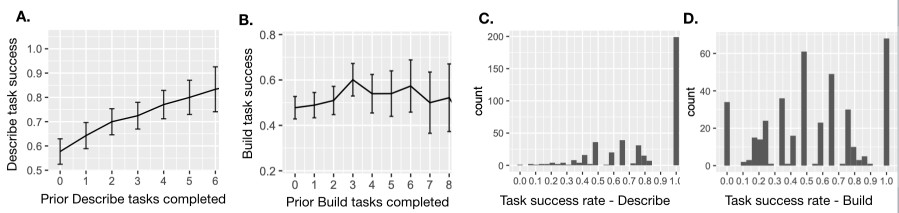

Figure 5: *A*. Describer improves at verifying their own descriptions as a they describe more tasks. *B*. Builders do not improve at constructing the correct outputs as they build more tasks (likely due to having no control over the qualities of their given descriptions). *C*. Rate of describers verifying their own descriptions (avg 75%). *D*. The rate of builders constructing the correct output, (avg 50%).

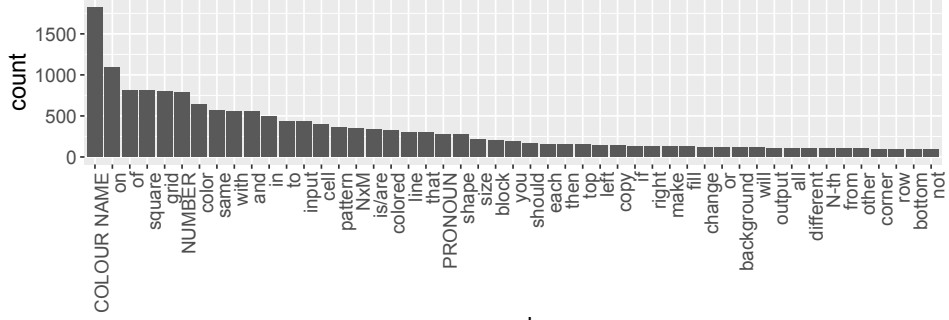

Figure 6: Words used in successfully built descriptions, sorted by their frequency in the corpus (total 642 unique words). The words were singularized. Colors names, numbers, and pronouns were grouped together.

*validated* by a number of builders, where both can make mistakes. Thus, A naive data-collection process that simply collects a fixed number of descriptions and builds per task will be expensive. To address this challenge, we formulate the following bandit problem: *multi-bandit* – each of the 400 ARC tasks is a different bandit; *infinite-arm* – given a task, each natural language description (there are infinitely many) is a different arm; *best-arm identification* – once a natural program is proposed, we must validate it. We develop a novel bandit algorithm (Appendix B) to solve this problem, as to our best knowledge, no known bandit algorithm can be directly applied. For each MTurk participant, our bandit algorithm dynamically allocates a set of describing and building efforts for their session. As a result, the LARC dataset was annotated for $3667, whereas a naively collecting 20 annotations per task would cost at least $10,800.

## 4 Communication Strategies in Natural Programs

What are some strategies humans use to produce robustly interpretable instructions? To answer this question, we curate a *linguistically tagged dataset* of tagged phrases from successful descriptions under the lens of *computer programs*. We annotate these phrases with *tags* corresponding to general concepts from algorithms and *core knowledge* [38]. In total, we manually label 532 randomly sampled phrases (22% of the phrase corpus) using 17 conceptual tags (in which multiple tags can be applied to each phrase); Figure 7A. shows a frequency of these tags. For details see Appendix A.3.

### 4.1 Similarities of Computer and Natural Programs

**General Algorithmic Concepts**    LARC contains algorithmic concepts similar to those found in a typical programming language (i.e. python). For instance, **tag_logic** is a boolean check (i.e. "the box is blue"), **tag_array** references a set of similar objects (i.e. "you should see four red shapes"), and **tag_loop** is similar to loops ("keep going until "). Humans generate these concepts without being directly instructed to do so, suggesting that humans reason about ARC tasks algorithmically.

**Domain Specific Concepts**    Similar to a computer DSL, LARC contains concepts that distinguish it from other domains. We focus on the object system of core knowledge [38], defined by *cohesion,*

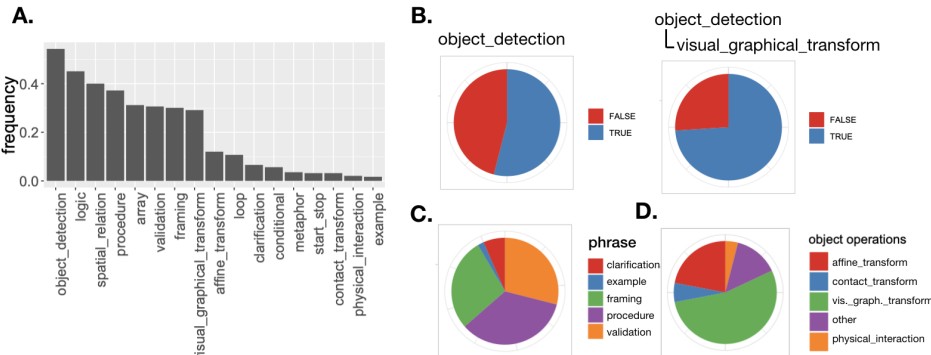

Figure 7: **A.** The frequencies of all tags occurring in human phrases. Each phrase can have multiple tags. **B.** More than half of the phrases described objects, of which, 75% described spatial relations. **C.** Relative frequencies of code (procedures) against non-code (example, framing, clarification, validation). **D.** Relative frequencies of core knowledge topics in phrases that referenced objects.

*persistence*, and *influence via contact*, which the ARC corpus was explicitly designed to leverage. We find about half of the phrases referenced **objects**, and three quarters of these described spatial relations (Fig. 7B). Majority of operations on objects (Fig. 7D) are **visual_graphical_transform** whereas only 5% of are **physical_interaction**. Presumably, graphical transformations are easier to represent in the input-output format of ARC.

## 4.2 Differences of Computer and Natural Programs

We outline two (related) ways natural programs differ from computer programs. First, instead of using a narrow DSL with few primitives, natural programs use a large, diverse set of primitive functions. Second, instead of stating a precise procedure verbatim, natural programs rely on a range of additional strategies to ensure that they can be interpreted precisely.

**Natural Programs Invoke a Large Number of Concepts**   Since LARC is language-complete, analyzing the words used in LARC serves as a good proxy for the underlying concepts present in the ARC domain. Similar to [21], we find that humans use a wide range of concepts (Fig 6). This is a testament of the general capabilities of the *human interpreter*: the describers readily invoke these concepts from the builders, with the confidence that they can be correctly interpreted. Given a large number of concepts, effectively indicating the set of relevant concepts (for a given task) becomes nontrivial: While human describers and builders can make use of generic word such as 'bump into', computer programmers must be extremely careful in selecting the exact concept using a precise language (i.e. `move_until_touches_block`).

**Natural Programs Communicate Information Beyond Procedures**   We study the relative frequencies of *directly executable* commands, **tag_procedure**, in contrast to not directly executable meta information such as **tag_framing** – comments about which concepts are relevant, **tag_validation** – checks to ensure correct execution, and **tag_clarifications** – restating the same procedure in different words. The most striking finding is that procedure, framing, and validation occur at roughly the same frequency (see Fig. 7 C). In contrast, only 14% of the codes are commented [39].

The high frequency of framing tags suggests that describers anticipate the large number of concepts that the builder can operate over, and carefully *frame* the instruction to invoke the appropriate ones. The describer often assumes the directly executable portion (i.e. tag_procedure) as inherently *ambiguous*, as suggested by frequent use of tag_validations and tag_clarifications following these procedures. Specifically, validation gives a check to the builder to test if their current interpretation is correct. Clarification amends the initial ambiguous explanation with another explanation, narrowing the number of possible interpretations. These are evidences that, unlike communication in computer programs over a *narrow and unambiguous* DSL, communication in natural programs are fundamentally *expressive yet ambiguous*, requiring extra efforts to maintain precision.

# 5   Executing Natural Programs using Program Synthesis

We evaluate whether current DSL-first program synthesis methods (Fig 3, bot) can execute natural programs as well as humans do. We consider three kinds of natural programs: (1) Input-output examples from the original ARC corpus (IO); (2) IO in conjunction with successful language instructions in LARC (IO+NL); And (3) language alone (NL-only) – same as the MTurk builder task.

## 5.1   Program Synthesis

In (symbolic) program synthesis [14, 19, 40], the synthesizer takes in a natural program, and reformulates it as code over a DSL to be executed. We have manually crafted a DSL based loosely on the concepts present in the LARC corpus and built its corresponding interpreter (see Appendix A.4) [6]. We present our best synthesis results here. For additional models (using a CNN encoder, a sequence decoder [19]) see A.5. Preliminary studies with *codex* and *clip* see A.6 and A.7.

**Generate and Check Using IO**   If the given natural program contains IO examples, the standard symbolic program synthesis approach [13, 14] follows the *generate and check* strategy. Let $natprog$ be a natural program, the synthesizer returns programs $prog$ from a DSL from the following distribution:

$$P_{synth}(prog|natprog) \propto P_{gen}(prog|natprog)\mathbb{1}[prog \vdash IO]$$

$P_{gen}$ is the generative distribution: given a natural program, it proposes program $prog$ from the DSL. $\mathbb{1}[prog \vdash IO]$ is the checker: it validates $prog$ by executing it on the interpreter, ensuring that $prog(x) = y$ for all input-output pairs $(x, y) \in IO$. The key strength of this approach lies in its generalizability: If a proposed program can be checked against all IO examples, it is very likely to generalize to an new instance of the same task due to the inductive bias of the DSL.

**Generation Models**   Our $P_{gen}(prog|natprog)$ generates programs in two parts: a neural model outputs a tree bigram over the grammar of the DSL [41], then a dedicated Ocaml enumerator deterministically enumerates programs from a probabilistic context free grammar fitted to this bigram distribution in decreasing probability [34]. For simplicity, we report results of unconditioned generators $P_{gen}(prog)$ (i.e. a fitted prior) when language is absent, and language-conditioned models $P_{gen}(prog|NL)$ when language is present. This way, we can use the same $P_{gen}(prog|NL)$ model for both IO+NL and NL-only tasks in the test set, as it does not depend on IO. Similar to [42, 43], we first bootstrap our generative models with 10 "seed" programs, discovered uninformed enumeration.

**Leveraging Language**   We use a pre-trained model (T5, [44]) to represent language by taking an average of its encoded tokens. To encourage the learning of compositional relationships between language and program, we use **pseudo-annotation**, similar to recent methods that have leveraged synchronous grammars [18, 33, 43, 45]. First, we provide linguistic comments for each primitive function in the program DSL (e.g. `flood_fill(color)` with *fill with the color*). Then, during training, we obtain additional paired language and program examples by substituting primitives of artificial programs with their corresponding comments [7]. For more examples see Appendix A.4.

**Distant Supervision**   LARC, similar to SCONE [46], falls under the challenge of distant supervision: each training task only contains the correct output, but not the ground-truth program responsible for generating it. We adopt the iterative approach used in [19, 34, 42, 43] to *discover* suitable programs during the training phase, by alternatively (1) generating a large sample of programs using $P_{gen}$ and (2) fitting a better $P_{gen}$ from good programs in the generated samples.

## 5.2   Results

We split the 400 tasks into 200 training tasks (with or without valid language descriptions) and 183 testing tasks (the remaining 200 filtered for having valid language deceptions). We then train the models for 10 hours each using iterative learning. We test on the 183 test tasks by first using the neural model to propose a bigram per task, then enumerating the bigram for 720 seconds. We keep

---

[6]this is a tremendous engineering effort, consisting of 103 primitives compared to 33 of SCONE
[7]for instance, `(lambda (to_original_grid_overlay (remove_color(grid_to_block x) yellow) false))` becomes *place block on input grid remove color from block yellow*

| training tasks discovered | | |
| --- | --- | --- |
| | no-pseudo | pseudo |
| IO | 15 / 200 | - |
| IO + NL | 13 / 200 | **21 / 200** |

| testing tasks solved | | |
| --- | --- | --- |
| | no-pseudo | pseudo |
| NL-only | 1 / 183 | 0 / 183 |
| IO | 18 / 183 | - |
| IO + NL | 16 / 183 | **22 / 183** |

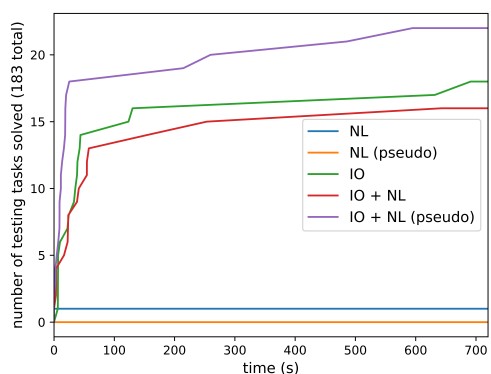

Table 1: Executing different kinds of natural programs (IO – Input-output examples from the original ARC corpus, IO+NL – IO in conjunction with successful language instructions in LARC, NL-only – same as the MTurk builder task) using program synthesis. Here, "pseudo" means the NL training has been pre-trained on generated synthetic language to code pairs. Train tasks discovered under distant supervision (top). Test tasks solved (bot).

Figure 8: Number of test tasks solved for the three kinds of natural programs, IO, NL, IO+NL, with and without pseudo annotations, as a function of enumeration time. There are error bars as the bigram enumerator is deterministic. It is possible (but not likely) that re-training these models will have an effect due to the randomness of sampling pseudo-annotated programs. All models vastly under performs when compared to a human, but natural programs consisting of NL+IO fairs best.

the top-3 most likely programs that also satisfy the IO examples if the natural program contains IO. We then check if any of the top 3 programs satisfies test input-output. See Table 1 and Figure 8. Overall, we conclude that while language definitely helps current approaches, the overall results (best 12%) are still comically bad.

**Quantitative Findings**  IO+NL+psuedo performs best, solving 22/183 of the testing tasks. We believe this due to psuedo-annotation being able to generate an infinite number (albeit low quality) of artificial NL-prog pairs. We note that having the ability to *check* if a proposed program is correct under IO is *crucial* for the success of current program synthesizers, with no more than 1 task solved with NL-only. Like the validation phrases in LARC, the input-output examples in IO serve as a form of *validation* for the enumerative synthesizer. This finding corroborates with [47].

**Qualitative Findings**  We investigate in what way does language affect synthesis. For each primitive in our DSL, we ask how many times more likely is it going to appear in correct programs generated with the language-conditioned bigram vs the unconditioned one. We plot this ratio on a log scale for all primitives that were used in ground-truth programs, see Figure 9. We note that for most of the frequently used primitives, the language-conditioned generator is more likely to generate the correct primitives than the unconditioned generator.

### 5.3  Challenges

The biggest challenge is **scoping**. Since LARC is DSL-open, we were in a vicious cycle of constantly adding more primitives and refactoring the DSL. Even now, we cannot guarantee our DSL can represent all LARC tasks. Second challenge is **referencing**: with 103 primitives, selecting the relevant primitives becomes crucial [8]. Finally, current NL-to-code approaches – like the ones we used – assume a close, 1-to-1 **paraphrase-like mapping** between language and procedure, which misinterpret crucial *framing* and *validation* statements that occurs in abundance in LARC.

---

[8]if we can magically select 10, the search space is $10^5$ instead of $103^5$ for a program of length 5

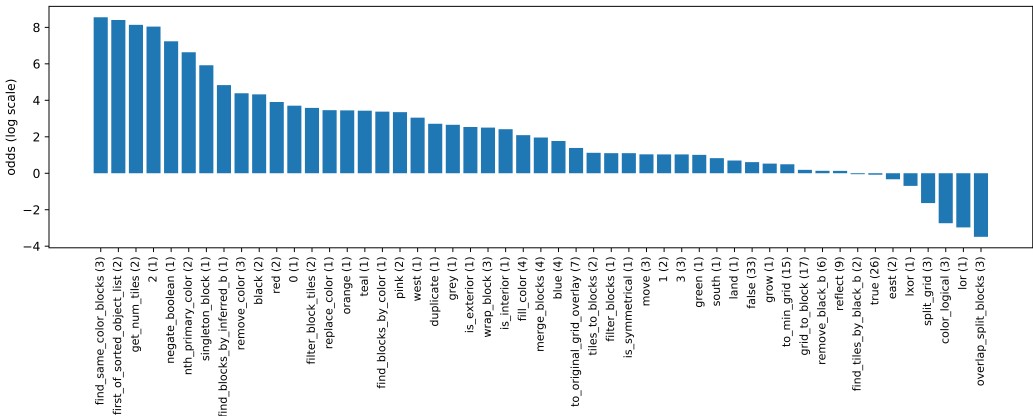

Figure 9: Relative odds of using the correct primitive for a task, language-conditioned vs unconditioned generation. Number in parenthesis denotes the total number of times a primitive is used.

# 6 Related Works

**Task oriented dialogue systems**    LARC as a dataset belongs to the family of task oriented dialogue systems [35–37, 48, 49]. One can view natural programs in LARC as a single-turn, task-oriented dialogue, where the describer gives a naturalistic instruction with a specific, check-able task in mind. Further, LARC uses the Wizard-of-Oz style of data collection – leveraging a human interpreter without committing to building a working system – a common framework to collect data in dialogue systems. LARC differs from these existing datasets mainly in the diversity of its tasks (Section 4) which contain a wide range of abstract concepts rather than being limited to specific domains such as database manipulations [37, 49].

**Embodied instruction following**    Embodied instruction following consisting of an embodied agent (often a avatar in a video game) being able to carry out a sequence of commands when prompted with natural language instructions [27, 50–53]. These commands can often be hierarchical [17, 50, 53], which are naturally represented as programs. LARC again differs from these works due to the range of abstract concepts, whereas aforementioned works typically follows a DSL-closed assumption.

As a result of a narrower range of concepts, a paraphrasal strategy that simply translate natural language into code has been fairly successful in prior works that aim to build an instruction following system [27, 49, 51]. LARC gives strong evidence that additional grounding strategies need to be modeled to truly capture the richness of natural language instructions (for instance, consider the set of strategies used in Fig 2).

# 7 Conclusion and Future Works

We present LARC, a *DSL-open* yet *Language-complete* dataset, highlighting the difference of between human-to-human and human-to-machines communications. By annotating successful communications (dataset of *linguistically-tagged-phrases*), we find that humans communicate using a wide range of concepts and communicative strategies, which are difficult to interpret using existing techniques. We hope LARC can help different communities (AI, Programming Language, Cognitive Science, etc) understand and build intelligent, communicative systems. Specifically, we believe that defining concepts upfront (DSL-first) is not scalable. Instead, they should be *learned* and *taught* (by end-users). To fully harness the power of natural language, we need to look beyond the simplistic notion that language having a 1-1 relationship with direct execution, and entertain different communicative strategies [54]. We believe datasets [51, 55, 56] that share the properties – namely, DSL-open and language-complete – are crucial to bridging the gaps between human-human and human-machine communications. Lastly, it will be beneficial to adapt foundational models [57–59] – with some conventional understandings of language, vision, and code – towards specific domains.

**Limitations and Potential Negative Impacts**  LARC consists of a single, constrained task format in a highly controlled setting. The long-term goal of this work is to 'reverse-engineer' how humans think and communicate, and such systems raise concerns regarding value alignments of users, for instance, non-experts operating safety-critical equipment using natural language.

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

## Acknowlegements

The authors would like to thank Eric Lu for inspiring the wonderful communication game that catalyzed our work.

