# OpenReview forum: "Communicating Natural Programs to Humans and Machines"
_NeurIPS.cc/2022/Track/Datasets_and_Benchmarks — NeurIPS 2022 Datasets and Benchmarks _

### Official Review · Reviewer_SWQ7 · 2022-07-21
**A natural language dataset for program synthesis**

**Rating:** 6
**Confidence:** 4

**Strengths:**

Strengths
- The paper addresses an important problem of generating programs from human natural language instructions. Specifically, the proposed dataset contain instructions that are complete for program synthesis, making them comparable to computer programs. In general, I believe such comparisons are essential for understanding the difference between human and machine intelligence, and for designing machines that can naturally interact with humans.
- The authors conduct sufficient analysis on the collected dataset, with additional annotation and data cleaning for quality assurance. I believe it is a valuable contribution to the program synthesis community.
- The dataset is also easily accessible, with an online user interface for visualization.
- The paper is generally well written. The authors provide sufficient details for readers to understand the contributions.


**Weaknesses:**

Weaknesses
- Although the task is more challenging, the size of the dataset is smaller than previous works (SCONE).
- Since the paper only concerns the grid-world domain which is rather toy-like and simplified, it is unclear that whether the conclusions can be applied to other domains.

**Additional Feedback:**

I appreciate the authors’ frankness for revealing the submission history. The previous responses can be helpful for evaluating the value of the paper.

**Clarity:**

The paper is generally well written. It can benefit from revisions for improved clarity:
- It seems some typos exist in lines 80, 92, 101.
- The abbreviation PCFG is not explained in the paper.


**Correctness:**

The paper includes both qualitative and quantitative evaluations for each task setting. The authors are recommended to address the following comments:
- It seems the equation above line 198 needs normalization so that LHS can be a valid probability distribution.
- A total of 383 tasks (200 for training and 183 for testing) are used in the experiment. In line 104, however, the number of tasks with verifiable natural program description is 354.

**Documentation:**

I believe the authors have released code and dataset with detailed instructions to reproduce the results.

**Ethics:**

There is no ethical issue from my perspective.

**Relation To Prior Work:**

I believe the paper discusses relevant works sufficiently. I'd suggest that moving supplementary table 3 to the main text to better visualize the contribution.

**Summary And Contributions:**

This paper looks at the problem of generating programs from human natural language instructions. The major contribution is a dataset with 354 verified human instructions that completely specify the procedure to generate output grids from the input for the ARC task. The collected human instructions are properly analyzed and compared with computer programs. Experiment results show that although these instructions can slightly improve the performance when training examples are presented, generating programs from natural instructions alone is still quite challenging.

I believe this is a good dataset paper. The paper itself is well motivated and well written. Despite the rather restricted domain and small scale, the dataset is properly validated and serves as a valuable contribution to the community. The comparisons between natural programs and computer programs can potentially inspire future works in this direction.

---

> ### Author Response · Authors · 2022-08-06
> **wonderful detailed feedbacks that can easily be integrated into the revision**
>
> ### LARC is small compared to Scone
> the smaller size of LARC is ultimately bounded by the smaller size of ARC tasks, which are manually curated by a single person rather than automatically generated as in SCONE. this problem of small size itself might motivate future research on how to crowd source creation of unique problem-solving tasks. Reviewer pgxu raised some really good related works that should be mentioned, we are (as of this response) working on including a proper related-work section (scone and others) to take advantage of the additional 10th page available for a revision.
>
> ### LARC is “toy-like”
> it is true that the appearance of the tasks are simple, but they contain complex abstract concepts such as sorting, analogy, assembly, and symmetry, which are applicable to real-world scenarios that would have a “higher fidelity in rendering and actuation”.  for instance, https://samacquaviva.com/LARC/explore/?task=100 is a task that involves replicating a pattern at a specific location and on a different scale (build a real building from a scaled model), https://samacquaviva.com/LARC/explore/?task=344 is a task that involves moving in a direction while avoiding an obstacle (bringing materials to the other side of the field as fast as possible).
>
> ### “the equation above line 198 needs normalization “
> great catch! we have replaced it with a \propto instead. thanks!!
>
> ### A total of 383 tasks
> This is badly worded indeed. the 200 training tasks include those w/o verified NL description (we need all the data we can), only the remaining 200 are filtered to 183 with verified NL descriptions. We have updated the manuscript.
>
> fixed the typos and abbreviations
>
> thanks again for such a detailed and constructive feedbacks!

---

> > ### Comment · Reviewer_SWQ7 · 2022-08-25
> > **Response to rebuttal**
> >
> > I appreciate the authors' response and revision. I have no further comment.

---

### Official Review · Reviewer_pGXu · 2022-07-23
**Adding natural language to ARC, considering the assumptions humans make on each vs on a machine**

**Rating:** 7
**Confidence:** 4
**Clarity:** The paper is clear.

**Strengths:**

There is a general shift in program synthesis to tackling higher level requests from users, and through learning from demonstration.  This work fits in this discussion. The key contribution is the re-use of ARC to use a human "builder" as a substitute to a machine, and then demonstrate that the resulting instructions provided by the "describer" may not look the same as when the describer knew they were communicating with a machine.

**Weaknesses:**

There is a lot of motivational language around “natural programs”, but without making the connection to the longstanding use of Wizard Of Oz protocols for the initial development of “human => interpreter” interfaces.  For example they write: “we suggest that starting with human instructions can yield insights for building systems”.  But haven't researchers been doing this since at least TRAINS and later TRIPS?  https://www.cs.rochester.edu/research/cisd/resources/trains.html

The focus is on an “open DSL”, but the assumption is still that you will map the language to some kind of representation via program synthesis. Does that make it open, or latent? I am uncertain whether this domain has an infinite set of useful predicates, or if there’d be a long tail of increasingly less important primitives, which at some point we consider the domain closed.

L168-171: While it is sufficient to recall an unique function name in writing a computer program, the set of names – ‘box’, ‘block’, ‘square’ – might mean same or different things under different contexts.
=> might these mean the same or different things under different contexts in programming languages? Perhaps more clear to say “fully resolved” function names?




**Additional Feedback:**


Minor aesthetics: the humans in Fig 3, I suggest having their arms raised at some angle above 0 degrees, it is distracting that they align with the arrows.

Is there a conflation between narrow vs open domains, as compared to programming languages vs natural language? You can have an expansive library in a programming language, and you can have narrow domain specific natural language. The protocol used in this work has one human communicating with another, who is a perfect stranger and without iterated interactions. The describer has to do everything ahead of time, without confidence in the capabilities of the builder.  As the authors make explicit, a human programer communicating with a computer can make more informed assumptions on the interpretive capacity of the “builder” (the machine). However, is the “natural” label more a spectrum, not a binary distinction? Under certain assumptions made by the describer on the builder, they will give a certain amount of details. Consider for example a military scenario, between a leader and a trusted subordinate: one expects communication in this case to be unambiguous and efficient.  This is in contrast to strangers on mechanical turk that never meet. Now consider a scenario where the describer was told that it would be a computer interpreter, using a certain programming language, and employing a certain library but where it wasn’t known which version of that library was available. The describer will be less efficient for lack of this knowledge, they will need to write more conditional code to verify which routines are available.

**Correctness:**

I would have preferred to see a Codex based result in the main body of the paper, but the experiments overall are reasonable.

The dataset appears interesting and I agree with the protocol, given the intent. For example it is a nice step that the describer was asked to perform the task themselves as a quality control.

**Documentation:**

Yes

**Ethics:**

"Subjects were paid $6.00 and a $0.25 112 bonus for every successful communication. Subjects averaged 5.5 communications, bringing their 113 expected hourly wage to $9.83."

MA has a minimum wage of $14.25/hr; consider more generous payments to crowd workers.

**Relation To Prior Work:**

TRAINS/TRIPS was discussed above.

More recently there is work like CerealBar which is worth making an explicit contrast with:

Executing Instructions in Situated Collaborative Interactions
https://aclanthology.org/D19-1218/


The SMCalFlow dataset supports a programmatic representation for clarifications, so a human can incrementally ask questions about domain state, and to correct or otherwise edit their own prior statements. Imagine asking the describer to schedule a meeting under various constraints, but done without the ability to know the result of incremental computations.  Would such a dataset have similar properties, but not be a human to human interaction?

Task-Oriented Dialogue as Dataflow Synthesis
SMCalFlow: https://aclanthology.org/2020.tacl-1.36/


This is one example of emerging work on driving agents in open worlds (here MineCraft) with language to code.  Would the authors still consider this human-machine interaction to be a limited DSL?

Craft an Iron Sword: Dynamically Generating Interactive Game Characters by Prompting Large Language Models Tuned on Code
https://aclanthology.org/2022.wordplay-1.3/




**Summary And Contributions:**

The authors add a natural language component to the ARCS challenge set. They propose the term “natural programs” as others would say “instructions”, to refer to human to human instructions are able to be accurately carried out and have a program execution-like result. The dataset will be interesting to those concerned with ARC and possibly will be of interest to those doing general program synthesis, but the positioning comes across as overly bold.

---

> ### Author Response · Authors · 2022-08-06
> **give us a few days to digest this !**
>
> we're updating a revision that address some of the other 5 reviewer's feedbacks.
>
> this set of feedback is very substantial, and we're currently in a paper reading spree trying to ingest this knowledge best we can, and will try our best to integrate these new set of related works into our manuscript. however this will take a few more days!
>
> but we just want to say thank you for this extremely helpful review, and made us want to know more about james allen's works more

---

> ### Author Response · Authors · 2022-08-08
> **a wonderful review that helped in further shaping the paper in a significant way**
>
> ### added a related work section
> with the extra 10th page, we were able to add a related work section that properly situates our work in context of prior works (including the ones you linked). for details see section 6 of the updated draft. to summarize, we believe LARC a good addition to this line of work because it documents and explicates the unique challenges in building a system in a DSL-open domain, containing a wide range of different concepts.
>
> ### “But haven't researchers been doing this since at least TRAINS”
> thanks for making the connection to the wizard of oz framework and the general topic of task oriented dialogue systems. these prior works are now acknowledged in the related work section and on line 85.
>
> ### on ARC DSL being open or latent
> we take DSL-open to mean [line 30] “– it does not come with a predefined DSL capable of solving all the tasks”, i.e. it is more of a property of a dataset, rather than if it is technically possible. we agree that under the definition of “DSL-closed := there exists a DSL capable of expressing all the tasks”, we can of course (with great pain) represent all 400 ARC tasks using OCaml. we have changed the definition slightly for better clarity “– it does not come with a predefined DSL capable of representing its tasks intuitively”.
>
> ### “ Perhaps more clear to say “fully resolved” function names?”.
> this is good point, the problem is more about “how to effectively invoke the right set of concepts at the right time” which is made very difficult under current programming environments when the semantics is only viewed as a proxy through either function name or doc-strings, at best some dynamic dispatch. we have changed to “Given a large number of concepts, effectively indicating the set of relevant concepts (for a given task) becomes nontrivial: While human describers and builders can make use of generic word such as `bump into', computer programmers must be extremely careful in selecting the exact concept using a precise language (i.e. \texttt{move\_until\_touches\_block}). “
>
> ### “is there a conflation between narrow vs open domains, as compared to programming languages vs natural language? … is the “natural” label more a spectrum, not a binary distinction?”
>
> the cases you brought up are good counter arguments, i.e. having a broad computer language (all of github), and a narrow natural language (stock traders circa 1980). this might just be an inaccuracy we have to left unaddressed given that in general, natural languages are more general and programming languages are more specific.
>
> ### Subjects were paid … every successful communication…”
>
> yes this is definitely an oversight on our end.
>
> we had originally intended to have an average of $12.00 / hr + additional bonus. this went awry in two following ways:
>
> we under-estimated the time of curating the LARC corpus (i.e. participants took more than 45 minutes for the batch of tasks the bandit allocated) – we implemented an algorithm that checks how long each task would take on average and to dynamically adjust the batch size, but by the time we got enough information to estimate this number more or less accurately, the experiment has already concluded.
> we also over-estimated the quality of the describer’s natural language descriptions (i.e. participants (both describers and builders) were not getting enough bonuses, which can only be retro-actively paid after the conclusion of the experiments).
>
> In the future we should be running smaller sessions of collections to get a better calibration of task difficulty, and starting with a more conservative estimate (i.e. 60 minutes as opposed to 45 minutes) so we can give fairer compensations.
>
> ### Minor aesthetics: the humans in Fig 3, I suggest having their arms raised at some angle above 0 degrees, it is distracting that they align with the arrows.
>
> ah ... no. this little person (in its current shape) is very sentimental to me! but I also agree with your feedback. we have modified the figure so that they do not align but rather just in parallel. hopefully that is a good compromise :]
>
>  again, thanks for such a detailed and informative response, it really made the paper better.

---

### Official Review · Reviewer_W6Rb · 2022-07-25
**Review of "Communicating Natural Programs to Humans and Machines"**

**Rating:** 8
**Confidence:** 4

**Strengths:**

1. This work is well-organized and performs a novel analysis of the intrinsic properties in the ARC benchmark, and proposes the LARC dataset to highlight these properties.
2. By introducing natural language descriptions collected from humans, the LARC problem is well-defined as a particular form of natural problems and program synthesis. The paper also gives a new and promising perspective on explaining the unique difficulty and how to solve ARC.
3. The paper have quantitative and qualitative findings comparing standard program synthesis and the natural program show how the human concept and communicative strategies differ from machines, calling for further study on leveraging language as an essential ingredient in program synthesis.

**Weaknesses:**

1. My major concern is that LARC's natural language descriptions explicitly instruct how to solve ARC tasks, which differs from the design of learning from input-output examples in the original ARC. The original ARC focuses on some crucial priors (e.g., object prior, goal-directedness prior, basic geometry, and topology priors) and the capability of reasoning and abstraction (i.e., learning from examples). From my perspective, the instructions drastically ease the task difficulty by bypassing the abstract process, instead of truly solving ARC problems (learn from examples vs. execute from instruction).
2. Adding natural language descriptions to the existing ARC dataset for program synthesis is good. However, given the existence of natural language descriptions and the rapid development of large program generative models (e.g., Codex, AlphaCode), I think the community would like to see how these models perform under natural language descriptions for program synthesis.

**Additional Feedback:**

I like the author's plan to adapt foundation models for LARC, since intuitively natural language descriptions can be better utilized through language models.

**Clarity:**

Yes, the paper is very well written. I enjoyed the flow of writing in the paper.

**Correctness:**

Yes, I think the claims made are good. The dataset collection process and evaluation methods are designed appropriately and performed correctly.

**Documentation:**

Yes, the documentation is good and the dataset is available.

**Ethics:**

I see no ethical concerns.

**Relation To Prior Work:**

Yes, I am not an expert in program synthesis, but from my perspective, the contributions and comparisons with previous works are well discussed in the paper.

**Summary And Contributions:**

The paper presents the LARC (Language-complete ARC) dataset containing natural language descriptions for how to solve ARC problems. These descriptions serve as ‘natural programs’ developed by human and is similar but different from traditional program synthesis (DSL-open vs. closed). Experiments show the difference between human-human and human-machine communication. This work shed light on introducing natural language (DSL-open problems) to program synthesis and communicative systems.

---

> ### Author Response · Authors · 2022-08-06
> **thanks for the feedbacks, looking forward to further discussions**
>
> ### on whether ARC or LARC is more challenging
>
> It isn’t always the case that ARC is more difficult than LARC tasks or vice versa. while the LARC tasks have “solution” all spelled out as language instructions, it is missing the visual queues of the example input-output, forcing the describers to verbalize these visual concepts using words alone, and forcing the builders to re-interpret / imagine these concepts.
>
> In practice we find that while most describers can solve the original ARC tasks, the descriptions they generate are often less reliable than the original input-output formats. for instance, https://samacquaviva.com/LARC/explore/description.html?task=272&id=1824857b-dcee-424d-a062-eb3ce27ff239 shows a describer capable of completing the original ARC task, yet their instruction can be mis-interpreted by some builders.
>
> the most easy set-up would be combining both visual IO along with natural language hints, and it would make an interesting study if we allow the describers to “trade” IO with NL to some degree, as certain concepts are easier to describe using words, and others using visual input-outputs. One could imagine a set-up where a person augment the LARC language-only descriptions with additional input-output images so that they can be more reliably interpreted, would these images resemble the original ARC images?
>
> ### on using foundational models
>
> our program synthesis model leverages T5 which is a large language model trained on language and code corpus. we have conducted a very preliminary study of using large language models (clip, codex/copilot) in appendix A6 and A7, the results suggest these models (as is) do not immediately understand LARC concepts. there needs to be a mechanism of teaching these foundational model additional, specialized concepts endemic to LARC in a data efficient way (i.e. not fine-tuning). if you have any ideas (even if they're just off the cuff) please let us know!
>
> thanks again for these interesting discussion points!

---

> > ### Comment · Reviewer_W6Rb · 2022-08-20
> > **Response to rebuttal**
> >
> > Thanks for your feedback. I think LARC is well-defined. I am sorry that I missed the appendix part of using foundation models. I acknowledge your preliminary trial with the codex model (though it might be better to utilize more common programming languages for the codex, e.g., python or java, instead of the few-shot copilot approach). That said, I still recommend acceptance for this work, and my score remains the same.

---

### Official Review · Reviewer_7UVZ · 2022-07-27
**Looks to be a valuable contribution to a hard problem**

**Rating:** 7
**Confidence:** 2
**Clarity:** The paper is clear and easy to read. …

**Strengths:**

The paper is well presented, motivated, and appears to provide a useful and accessible resource. The explanation of the motivation, data collection, and characterization of the data is comprehensive. The supplementary material is also thorough.

**Weaknesses:**

I did not notice any limitations in the dataset (although I will not claim to be an expert in this domain).

**Additional Feedback:**

None currently.

**Correctness:**

The dataset collection approach appears to be reasonable, especially considering cost and practical issues.

**Documentation:**

The documentation appears comprehensive. I noticed that the supplementary material includes a Datasheet for the dataset. I'm not sure if this is available on the github repo, so it would be good to have that there (perhaps linked in the Readme?)

**Ethics:**

The author mentions that IRB approval was sought and received.

**Relation To Prior Work:**

The motivation of the work is well explained. There does not appear to be much in the way of comparing to other "natural program" datasets, but I am not aware of any other that exists.

**Summary And Contributions:**

This paper presents an enhancement to the Abstraction and Reasoning Corpus by adding human-written "natural programs" sourced via Amazon Mechanical Turk. The contributions include the dataset, insights into how the data was collected, and results of an initial experiment of program synthesis using the data.

---

> ### Author Response · Authors · 2022-08-06
> **thanks for the feedbacks**
>
> thank you for the positive reception, and we're glad it was easy to follow for someone outside the domain.
>
> there are a few other datasets that might be considered “open-dsl”:
>
> https://arxiv.org/abs/2106.14321 where the participants draw a figure based on natural language descriptions
>
> https://aclanthology.org/P19-1537/ where an architect communicate with a builder to construct a scene in minecraft
>
> both of these datasets are not expressly framed as a program synthesis dataset, but they are indeed relevant. we have added these references to the conclusion section of our paper, with a comment that we ought to collect more datasets like them
>
> added datasheet pdf under LARC/datasets
>
> edit: reviewer pgxu has raised some really cool related works that can be considered "open-dsl" as well, we are working on integrating these information into the main paper.

---

### Official Review · Reviewer_W2sg · 2022-07-27
**Very interesting and solid, down-stream re-use potentially limited**

**Rating:** 7
**Confidence:** 2
**Correctness:** All claims seem correct to me; method…
**Clarity:** The paper is written very clearly.

**Strengths:**

The crowdsourcing setup used for creating the dataset is solid and very elegant. The dataset itself is relevant for its field and a useful extension of the established ARC dataset. All reported work is very solid and of high quality.

**Weaknesses:**

I am sceptical about the versatility of the dataset for enabling extensive down-stream re-use by other research groups. This seems relevant given that the paper was submitted to the Datasets track, and I think that other submitted datasets have a stronger potential for wide re-use.


**Additional Feedback:**

N/A

**Documentation:**

I could not find a link to the actual dataset in the main manuscript and found the corresponding Github page via Google. Unless this is a major oversight on my side, please add the links to the manuscript!

The manuscript provides all necessary details on data collection and organization, availability and maintenance, and ethical and responsible use.


**Ethics:**

There are no ethical concerns.

**Relation To Prior Work:**

Prior work is sufficiently discussed.

**Summary And Contributions:**

The authors provide an interesting natural language extension to the pre-existing Abstraction and Reasoning Corpus (ARC) dataset. They created natural language instructions that provably were able to instruct humans to solve ARC tasks in a sophisticaed crowdsourcing setup.

The authors provide extensive work around characterizing the resulting natural lanaguage instructions, as well as applying domain specific languages (DSL). The paper focuses heavily on discussion of DSLs, and my expertise is unfortunately very limited in this domain. To me it seems like the findings mostly point to the limitations of DSLs.

Beyond DSLs, the dataset might also be useful for experimenting with solving the ARC tasks with large language models.

---

> ### Author Response · Authors · 2022-08-06
> **thanks for the positive feedbacks and suggestions**
>
> we’re so glad you enjoyed the bandit algorithm! It was such a deep nerd dive down the bandit route for a few months. a key insight really is “best arm identification” boils down to “reducing entropy about p* of a casino”, and we’ve done a version of the algorithm where this information is computed using a bayesian inference which ultimately was too slow to run on a 400-task set up, before finally settling for our cheaper approximation. it is good to see these efforts are being appreciated!
>
> the usability of down-stream tasks is definitely a concern. for us, this dataset serves both as an open challenge as well as quantifying (and being explicit) about what the challenges are (scoping, dsl-open vs dsl-closed, communicative strategies) so they can be tackled individually. we believe as the field of ML advances, it will become more accessible to downstream tasks
>
> we have added the link to the LARC github to the main document

---

> > ### Comment · Reviewer_W2sg · 2022-08-24
> > **Thanks**
> >
> > Thanks for adding the link! I have no further comments.

---

### Official Review · Reviewer_Veu8 · 2022-07-28
**A nice dataset to spur further research in program synthesis**

**Rating:** 8
**Confidence:** 4
**Clarity:** The paper is very well-written and ea…

**Strengths:**

* The paper tackles an important problem and the dataset provides a useful contribution to fuel advances in AI research.
* Natural programs are a very interesting perspective that can inspire the design of novel approaches to program synthesis for DSL-open domains.
* The dataset is well curated and easy to explore.
* Experimental results show how challenging the task is and provide a useful set of baselines (finally) that will hopefully encourage more researchers to work on the ARC tasks.


**Weaknesses:**

* The paper is very well-written but the presentation of the work could benefit from some changes. For instance, it is not immediately clear what some tags in Figure 7 refer to and it would be helpful if the main paper could provide more details on the baselines.
* It would be useful if the main paper could provide a link to the dataset and the source code to replicate the experiments (it seems the Github repository does not include the code to run the baselines on the dataset)

Minor comments:
* It would be useful to have the ARC task ID in Figures 1 and 2 instead of a numeric index
* Line 24: I disagree that ARC (by itself) presents tasks designed to benchmark the capacity to communicate.

**Additional Feedback:**

None

**Correctness:**

The claims of the paper are correct and the dataset is constructed in a sound way

**Documentation:**

The paper provides details on how the data was collected and the dataset is easy to explore

**Ethics:**

No ethical concerns

**Relation To Prior Work:**

The paper clearly discusses relation to prior work

**Summary And Contributions:**

The paper presents an extension of the ARC dataset where tasks are labeled with natural-language descriptions that are sufficient for a human to generate the output grid, given an input grid without any additional examples demonstrating the task. The paper further provides annotations on the natural-language descriptions, showing how humans communicate natural programs and which concepts are needed for the ARC tasks.

---

> ### Author Response · Authors · 2022-08-06
> **thanks for the positive reception and feedbacks, we have made all the suggested changes**
>
> “it is not immediately clear what some tags in Figure 7 refer to and it would be helpful if the main paper could provide more details on the baselines” (Tagging details are in appendix A.3 and baseline details are in A.4, A.5. A.6. and A.7. We agree these details are useful but the space of 9 pages, ah … )
>
> “It would be useful if the main paper could provide a link to the dataset and the source code” (added larc github to main paper, the synthesis link is in that repo, under “contents”)
>
> “It would be useful to have the ARC task ID in Figures 1 and 2 instead of a numeric index” (done)
>
> “Line 24: I disagree that ARC (by itself) presents tasks designed to benchmark the capacity to communicate” (we agree that this is misleading, the original intent is more along the line of “self communication” where you summarize the rules for the ARC tasks to your future self when applying it on a new input. We have removed this since it is confusing).
>
> again thanks for taking the time to review this paper and catching these issues!

---

> > ### Comment · Reviewer_Veu8 · 2022-08-25
> > **Satisfied with the author reply**
> >
> > I thank the authors for their reply, which addresses all my concerns. I am satisfied with the rebuttal and I increased my score by 1 point.

---

### Meta-Review · Area_Chair_P7N6 · 2022-09-09

**Recommendation:** Accept
**Confidence:** 5

**Metareview:**

The paper addresses the important problem of program synthesis from human natural language instructions. Sufficient analysis is done on the collected dataset, including annotation and data cleaning for quality assurance and the work is a valuable contribution to the program synthesis community.  Based on the reviews, I recommend acceptance of the paper.

---

### Decision · Program_Chairs · 2022-09-16

Accept